# Cry Toxins Use Multiple ATP-Binding Cassette Transporter Subfamily C Members as Low-Efficiency Receptors in *Bombyx mori*

**DOI:** 10.3390/biom14030271

**Published:** 2024-02-23

**Authors:** Satomi Adegawa, Yonghao Wang, Ryusei Waizumi, Tetsuya Iizuka, Yoko Takasu, Kenji Watanabe, Ryoichi Sato

**Affiliations:** 1Graduate School of Bio-Applications and Systems Engineering, Tokyo University of Agriculture and Technology, 2-24-16 Koganei, Koganei 184-8588, Tokyo, Japan; adegawa-satomi@hro.or.jp (S.A.); hobbswang0503@gmail.com (Y.W.); 2Japan Society for the Promotion of Science Research Fellowship for Young Scientists, 5-3-1 Kojimachi, Chiyoda-ku 102-0083, Tokyo, Japan; 3Institute of Agrobiological Sciences, NARO, 1-2 Ohwashi, Tsukuba 305-8634, Ibaraki, Japan; waizumir957@affrc.go.jp (R.W.); tiizuka@affrc.go.jp (T.I.); takasu@affrc.go.jp (Y.T.)

**Keywords:** cry toxin, ABC transporter, SPR analysis, low-efficiency receptor, binding affinity

## Abstract

Recent studies have suggested that ABC transporters are the main receptors of Cry toxins. However, the receptors of many Cry toxins have not been identified. In this study, we used a heterologous cell expression system to identify *Bombyx mori* ABC transporter subfamily C members (BmABCCs) that function as receptors for five Cry toxins active in Lepidopteran insects: Cry1Aa, Cry1Ca, Cry1Da, Cry8Ca, and Cry9Aa. All five Cry toxins can use multiple ABCCs as low-efficiency receptors, which induce cytotoxicity only at high concentrations. Surface plasmon resonance analysis revealed that the *K_D_* values between the toxins and BmABCC1 and BmABCC4 were 10^−5^ to 10^−9^ M, suggesting binding affinities 8- to 10,000-fold lower than those between Cry1Aa and BmABCC2, which are susceptibility-determining receptors for Cry1Aa. Bioassays in BmABCC-knockout silkworm strains showed that these low-efficiency receptors are not involved in sensitivity to Cry toxins. The findings suggest that each family of Cry toxins uses multiple BmABCCs as low-efficiency receptors in the insect midgut based on the promiscuous binding of their receptor-binding regions. Each Cry toxin seems to have evolved to utilize one or several ABC transporters as susceptibility-determining receptors.

## 1. Introduction

Cry toxins, crystal proteins produced by the soil bacterium *Bacillus thuringiensis*, have been classified into several families. In 2021, Crickmore et al. proposed that only toxin proteins with a three-domain structure should be termed Cry toxins [1]; we follow this paradigm. There are diverse genotypes of Cry toxins. Each Cry toxin is toxic to a limited range of insect species; thus, they exhibit a narrow insecticidal spectrum. Because of their minimal toxicity to vertebrates such as mammals and fish, Cry toxins are used worldwide in agriculture for pest control. Mechanistic analyses support classifying Cry toxins as pore-forming toxins; such toxins bind to target cells, cluster together, and pierce the plasma membrane, creating a small pore [2]. The pores created by Cry toxins allow the passage of ions such as K^+^ and Ca^2+^, thereby increasing intracellular osmotic pressure [3]. Subsequently, water flows into the cell, mainly via the water channel aquaporin, leading to cell swelling and rupture [4]. The receptors for Cry toxins are proteins (e.g., adenosine triphosphate-binding cassette [ABC] transporters) that not only bind to Cry toxins with high affinity but also promote pore formation in the plasma membrane [5].

In Lepidopteran insects, mutations in ABC transporter subfamily C member 2 (ABCC2) result in resistance to Cry1A toxins [6,7,8,9,10]. ABC transporters are divided into eight subfamilies: ABCA to ABCH [11]. Among the 760 diverse genotypes of Cry toxins, the phylogenetically separate Cry1A and Cry2A, as well as the intermediate lineage Cry3A, use ABCC2, ABC transporter subfamily A member 2 (ABCA2), and ABC transporter subfamily B member 1 (ABCB1) as their respective receptors [6,12,13,14,15]. Recently, we showed that Cry1Ba, Cry1Ia, and Cry9Da toxins also use ABCB1 as a receptor [16]. However, studies of receptors of other Cry toxins have made little progress. For example, the susceptibility of Cry1Aa toxin-resistant insect strains to Cry1Ca and Cry1Da suggests that Cry1Ca and Cry1Da do not use ABCC2 [17]. The receptors for Cry1Ca and Cry1Da are unknown. On the other hand, Cry1Aa, Cry1Ab, Cry1Ac, and Cry1Fa use both ABCC2 and ABC transporter subfamily C member 3 (ABCC3) as receptors [10,18,19,20]. This phenomenon implies that a single Cry toxin can use multiple ABC transporters as receptors. Furthermore, binding affinity analysis via surface plasmon resonance (SPR) and cell swelling assays with cultured cells ectopically expressing ABC transporters have indicated a correlation between the ABC transporter-binding affinities of Cry toxins and the Cry toxin susceptibilities of ABC transporter-expressing cells [19,20,21,22]. Therefore, it is becoming increasingly evident that Cry toxins exhibit high-affinity binding with ABC transporters. However, the evolutionary trajectories of Cry toxins, specifically how they achieved high-affinity binding to ABC transporters, are unclear. Insights regarding these issues require the identification of Cry toxins that use particular ABC transporters as receptors.

In this study, we identified *B. mori* ABC transporter subfamily C members (BmABCCs) that serve as receptors for Cry1Aa, Cry1Ca, 1Da, 8Ca, and Cry9Aa. A cell swelling assay using a heterologous expression system in HEK293T or Sf9 cells revealed that five Cry toxins use multiple BmABCCs as receptors. However, the swelling of BmABCC-expressing cells required toxin concentrations >500-fold higher than the concentration of Cry1Aa required for cultured cells expressing BmABCC2. Subsequently, our SPR analysis indicated *K_D_* values 8- to 10,000-fold higher than the values for binding between Cry1Aa and BmABCC2. Furthermore, we conducted bioassays using *B. mori* strains with truncated BmABCCs. The results show that Cry toxins have multiple low-efficiency receptors that do not significantly contribute to the susceptibility of silkworm larvae to Cry toxins. We proposed a hypothesis as to how Cry toxins evolved to use particular ABC transporters as highly efficient receptors and explored contributing factors.

## 2. Materials and Methods

### 2.1. Preparation of Three-Domain Cry Toxins

We prepared Cry1Aa, Cry1Da, Cry8Ca, and Cry9Aa protoxins as recombinant protein from *Escherichia coli* as described previously [21,23,24,25,26]. The Cry1Ca protoxin was produced from a *B. thuringiensis* recombinant strain as described previously [21]. These protoxins were activated by trypsin and purified to be activated Cry toxins as described before [24,26].

### 2.2. Analysis of B. mori Midgut RNA-Seq Data

We downloaded *B. mori* midgut RNA-seq data (SRR1806736) from NCBI and removed fragments with a quality below 20 and a base length below 20 bp using the FASTX-toolkit program (http://hannonlab.cshl.edu/fastx_toolkit/ accessed on 20 December 2022). Next, we downloaded the nucleotide data of *B. mori* from the NCBI website and used them as a reference for mapping RNA-seq reads. We used Bowtie 2 (http://bowtie-bio.sourceforge.net/bowtie2/index.shtml accessed on 20 December 2022), a mapping program which maps short reads of 50–100 bp to a reference, and mapped them to the prepared *B. mori* reference data. A total of 83.12% of all reads were mapped. The mapping result and read count data were submitted to eXpress (https://pachterlab.github.io/eXpress/index.html accessed on 20 December 2022), a program which quantifies and displays gene read count data, to compare the expression levels of genes. In eXpress, the expression level can be evaluated by the transcripts per million (TPM). Finally, the accession numbers registered as *B. mori* ABC transporters in NCBI were extracted from the eXpress results and arranged in order of their increasing expression levels.

### 2.3. cDNA Cloning of B. mori ABCC Transporters

Total RNA from midgut tissues of *B. mori* was isolated and used for cDNA synthesis as described before [21]. The cDNAs of BmABCC1 (XM_012690682.3), BmABCC4 (XM_021349664.2), BmABCC10 (XM_004925521.4), and BmABCC11 (XM_021350972.1) were amplified by PCR using each primer shown in Appendix A and the cDNA of *B. mori* midgut as the templates. The amplified cDNAs were cloned into the EcoRV site of the pcDNA3.1 vector (Thermo Fisher Scientific, Tokyo, Japan) as described previously [21].

### 2.4. Preparation of AcNPVs for the Expression of BmABCC1 and BmABCC4

Recombinant Autographa californica nuclear polyhedrosis virus (AcNPV) for the expression of BmABCC1-FLAG and BmABCC4-FLAG was produced as described previously [24].

### 2.5. Cell Swelling Assay

Human embryonic kidney (HEK) 293T cells were cultured and used for transfection as described previously [27]. A cell swelling assay using transfected HEK293T cells was performed as described before [21]. Briefly, HEK 293T cells transfected with each of the six BmABCCs (BmABCC2, BmABCC3, BmABCC4, BmABCC10, BmABCC11) were incubated with five activated Cry toxins (Cry1Aa, Cry1Ca, Cry1Da, Cry8Ca, and Cry9Aa). After 60 min of incubation in a CO_2_ incubator at 37 °C, the cells were observed under a phase-contrast microscope. Cell swelling assays using recombinant AcNPV-infected Sf9 cells were performed and analyzed as described previously [26]. Briefly, Sf9 cells expressing BmABCC1 or BmABCC4 were incubated with four Cry toxins (Cry1Aa, Cry1Da, Cry8Ca, and Cry9Aa), respectively. After 60 min of incubation at room temperature, the cells were observed under a phase-contrast microscope.

### 2.6. Affinity Purification of BmABCC1 and BmABCC4

BmABCC1-FLAG and BmABCC4-FLAG was expressed in the Silkworm-Baculovirus System produced by ProCube (Sysmex Corporation, Hyogo, Japan). BmABCC1-FLAG and BmABCC4-FLAG cDNAs were inserted in the pHS02 transfer vector (Sysmex Corporation). This transfer vector was co-transfected with BmNPV DNA [28] into a *B. mori*-derived cell line (BmN) [29]. After 7 days of incubation, the recombinant baculovirus was injected into the hemocoel of silkworm pupae. The pupae were harvested 6 days after infection, frozen, and homogenized with PBS, 1 tablet/50 mL protease inhibitor cocktail, and 25 mg/50 mL phenylthiourea. The homogenized solutions of infected pupa were suspended and solubilized in 1% n-dodecyl-β-D-maltoside (DDM) (Dojindo, Tokyo, Japan) for 60 min at 4 °C. After incubation, the solution was centrifuged at 24,000× *g* for 40 min at 4 °C, and solubilized BmABCC1-FLAG and BmABCC4-FLAG were obtained as supernatants. Then, they were purified and prepared for SPR analysis as described previously [26]. The purity of BmABCC1 and BmABCC4 was shown in Appendix A.

### 2.7. SPR Analysis (BIACORE)

The buffer of the BmABCC1 and BmABCC4 solutions was changed to PBST (PBS containing 0.005% Tween 20) and immobilized on a CM5 sensor chip as described in Adegawa et al., in 2017 [26]. Cry1Aa, Cry1Da, Cry8Ca, and Cry9Aa toxins were diluted in PBST buffer. Cry1Aa and 1Da were applied to the BmABCC1-immobilized sensor chip for 120 s at 1000 nM; Cry8Ca and Cry9Aa were applied at 500 and 1000 nM. For the sensor chip immobilized with BmABCC4, Cry1Aa, 1Da, and 8Ca were applied at 500 and 1000 nM, and Cry9Aa was applied at 1000 nM. Measurement and analysis of binding kinetics and affinity were performed as described previously [30].

### 2.8. Silkworm Strains

The wild-type silkworm strains w-1 and Ringetsu were distributed from the Genetic Resources Center, National Agriculture and Food Research Organization (NARO), and were reared on mulberry leaves or an artificial diet (Nihon Nosan Kogyo, Yokohama, Japan) at 25 °C. The genome-edited strains of BmABCC1 (Appendix A) and BmABCC4 (Appendix A) were generated by microinjection of TALEN mRNAs according to a previous report [15]. The TALEN target sites were selected in the extracellular loop4 (ECL4)-encoding region in the 16th exon of the *BmABCC1* gene and the transmembrane 1 (TM1)-encoding region in the 3rd exon of the *BmABCC4* (Appendix A). To identify mutations induced by TALENs, G1 individuals were obtained by crossing injected individuals (G0) and wild-type strains, and the PCR products of the target regions were amplified by specific primer sets (Appendix A, BmABCC1_geno, or BmABCC4_geno) and sequenced as described previously [15]. The G1 individuals detected to carry the same mutations were crossed in single pairs to generate G2 families. G2 individuals were screened, and those identified to be homozygous for the same mutant alleles were pooled to establish mutant strains. To determine the ECL4 region of BmABCC1, the amino acid sequences of BmABCC2 and BmABCC1 were aligned (Appendix A) based on the topology of BmABCC2 predicted using the online software Polyphobius (http://phobius.sbc.su.se/, accessed on 10 December 2023), with a slight modification to the prediction by Tanaka et al. [31] (Appendix A). And, the amino acid sequence of extracellular loop 4 was aligned using the multiple alignment system of the CLC Sequence Viewer (QIAGEN, Venlo, The Netherlands). Based on the result of the alignment, the ECL4 region of BmABCC1 is located from the 954th to the 1003rd amino acid residues (Appendix A).

### 2.9. Diet Overlay Assays

The prepared protoxins of Cry1Aa (AAA22353), Cry1Da (CAA38099), Cry8Ca (AAA21119), and Cry9Aa (CAA41122) were used in the diet overlay bioassay. Susceptibility to the Cry1Aa, Cry1Da, Cry8Ca, and Cry9Aa toxins of genome-edited *B. mori* strains carrying one receptor gene among BmABCC1 or BmABCC4 as well as the wild type (w-1 or Ringetsu, respectively) of the strain was evaluated with diet overlay bioassays. Leaf disks were picked up that were the sixth open leaves from the top of each branch of the mulberry trees. The Cry1Aa, Cry1Da, Cry8Ca, and Cry9Aa protoxin solutions were diluted with Silwet^®^L-77 (Momentive Performance Materials, Waterford, NY, USA), and the protoxins were spread to be 10 μL/cm^2^ thick on the leaf disks. When the protoxin solutions were dried at room temperature, two leaf disks were put in one Petri dish with ten *B. mori* larvae of the second instar of the wild type and BmABCC1 or BmABCC4 genome-edited strains. Two days after administration, the leaf disks contaminated with toxins were replaced with fresh ones. The larvae were reared for a total of 4 days, and the mortality was recorded. Then, the median lethal dose (LC50) values and the 95% confidence interval were calculated based on a probit Analysis [15].

## 3. Results

### 3.1. Selection of B. mori ABC Transporters

As a preliminary step to identify ABC transporters that serve as receptors for Cry1Aa, Cry1Ca, Cry1Da, Cry8Ca, and Cry9Aa, we first investigated the ABC transporters of the silkworm moth *B. mori*, which is a model organism.

Xie et al. identified 53 ABC transporters in the *B. mori* genome [32]. However, their sequence analyses were based on SilkBase data prior to database revision. Furthermore, the names of the *B. mori* ABC transporters in the NCBI database lacked organization and were inconsistent. Therefore, we cross-referenced the *B. mori* ABC transporter information in the NCBI database with the results of a genomic analysis of silkworm ABC transporters [32]. We conducted a BLAST analysis to reorganize and rename 50 ABC transporters of *B. mori* by eliminating duplicates (Appendix A). In this study, for the ABC transporters who had already been reported to be receptors for Cry toxins, we adopted the names provided by each Cry toxin research group (ABCA2, ABCB1, ABCC2, and ABCC3). For the rest, we basically followed the registration names on the NCBI (ABCC1, ABCC4, ABCC10, ABCC11, etc.). However, for those not labeled with “ABC transporter” on the NCBI (uncharacterized LOC101740576 or mitochondrial potassium channel, etc.) and for those with duplicate names on the NCBI (MRP49 or ABCG1, etc.), we assigned names based on the results of our BLAST analysis, using the same names as the closest *Homo sapiens* ABC transporter (Table 1 and Appendix A). We also analyzed the sequence using InterPro, a web-based protein domain prediction tool (https://www.ebi.ac.uk/interpro/ accessed on 20 January 2024), and confirmed that all the sequences are ABC transporter family proteins containing the Walker A, B, and ABC signature C motifs (Appendix A). Next, we analyzed RNA-Seq data from the midgut of *B. mori* (SRR1806736); these data are publicly available in the NCBI Sequence Read Archive (SRA). Our analysis revealed that 46 ABC transporters are expressed in the midgut of *B. mori* (Appendix A). The HEK293T cell assays in previous reports suggested that ABCC2, 3, and 4 function as receptors for Cry1 or Cry8 [20,21]. We speculated that other members of the ABC transporter subfamily C may also be receptors for Cry toxins. Therefore, we investigated members of the ABC transporter subfamily C in *B. mori* (BmABCCs).

For our phylogenetic analysis, we selected 58 sequences, including the sequences of 10 BmABCCs in the NCBI database and ABCCs from *H. sapiens*, *Drosophila melanogaster*, and *Helicoverpa armigera*. The ABC transporter subfamily A member 1 from *H. sapiens* (HsABCA1) was included as an outgroup. Using HsABCCs as reference sequences, the 10 ABC transporter subfamily C molecules from *B. mori* were classified into the HsABCC4, HsABCC10, HsABCC2, and HsABCC8/9 clades (Appendix A). In the silkworm midgut, the 10 BmABCCs had transcripts-per-million (TPM) values in the order ABCC2, ABCC4, ABCC3, ABCC10, ABCC11, ABCC1, ABCC4-like1, ABCC9, ABCC4-like2, and ABCC5 (Appendix A). We cloned four cDNAs: BmABCC1 (XM_012690682.3; HsABCC2 clade), BmABCC10 (XM_004925521.4; HsABCC10 clade), and BmABCC4 and BmABCC11 (XM_021349664.2 and XM_021350972.1; HsABCC4 clade), in addition to BmABCC2 and BmABCC3 (HsABCC4 clade) [21]. In the predicted topologies, BmABCC2, BmABCC3, BmABCC4, and BmABCC11 had six ECLs and twelve transmembrane helixes, whereas BmABCC1 and BmABCC10 had five additional transmembrane helixes on the N-terminal side (Figure 1).

### 3.2. Receptor Functions of BmABCCs

To analyze whether BmABCC1, BmABCC2, BmABCC3, BmABCC4, BmABCC10, and BmABCC11 function as receptors for Cry toxins, HEK293T cells expressing the six BmABCCs were treated with >1000 nM of Cry1Aa, Cry1Ca, Cry1Da, Cry8Ca, or Cry9Aa; cell swelling was observed. The expression levels of the BmABCCs in the HEK293T cells were confirmed using enhanced green fluorescent protein (EGFP), which was fused to the C-terminus of each BmABCC. The percentage of fluorescent cells was approximately 40% of the total cells observed (Appendix A). The cell swelling assay results are presented in Figure 2. The swollen cells are indicated by arrowheads; the darker regions are the result of changes in refractivity during cell membrane detachment from the cytoplasm (Figure 2). Cry1Ca and Cry9Aa caused swelling in BmABCC1-expressing cells, whereas Cry1Aa and Cry8Ca caused swelling in BmABCC2-expressing cells. Cry1Aa and Cry9Aa caused swelling in BmABCC3-expressing cells; Cry1Aa, Cry1Da, and Cry8Ca caused swelling in BmABCC4-expressing cells. Cry1Ca, Cry1Da, and Cry9Aa caused swelling in BmABCC10-expressing cells, while Cry1Da and Cry9Aa caused swelling in BmABCC11-expressing cells. The percentage of swollen cells was low in all cases; however, multiple swollen cells were found when the microscope’s field of view was moved. These results suggest functional relationships between the six BmABCCs and the five Cry toxins, whereby BmABCCs function as receptors for Cry toxins. The other combinations of BmABCCs and Cry toxins did not cause cell swelling. HEK293T cells expressing EGFP alone did not cause cell swelling in response to any of the five Cry toxins (Figure 2).

### 3.3. Sensitivities of Sf9 Cells Expressing BmABCC1 or BmABCC4 to Cry1Aa, Cry1Da, Cry8Ca, and Cry9Aa

The sensitivity of cell swelling assays is more than 100-fold higher when using Sf9 cells than when using HEK293T cells, likely because Sf9 cells exhibit higher expression levels of ABC transporters. This difference is demonstrated by the varying toxin concentrations required to induce cell swelling when using HEK293T cells versus Sf9 cells for the same combination of BmABCC2 and Cry1Aa or Cry1Ac. [15,24]. Therefore, to validate the results obtained with transfected HEK293T cells, we created recombinant baculoviruses expressing BmABCC1 or BmABCC4 and then used those viruses to infect Sf9 cells. The infected cells were subjected to cell swelling assays using Cry1Aa, Cry1Da, Cry8Ca, and Cry9Aa. Cry1Ca was not used because Sf9 cells are sensitive to it. First, we confirmed the expression levels of BmABCC1 and BmABCC4 in the Sf9 cells through the co-expression of EGFP. The expression levels of recombinant baculoviruses expressing BmABCC1, BmABCC2, BmABCC4, or EGFP alone were 67.5%, 81.8%, 88.5%, and 84.2%, respectively (Appendix A). The BmABCC1-expressing Sf9 cells swelled when exposed to >500 nM Cry9Aa (Figure 3A), with 14% of cells swelling when exposed to 5000 nM Cry9Aa. The percentage of swollen cells increased in a dose-dependent manner (Figure 3B). However, Cry1Aa, Cry1Da, and Cry8Ca at 5000 nM did not cause cell swelling (Figure 3A). On the other hand, among the BmABCC2-expressing Sf9 cells, 90% of cells were swollen by 10 nM Cry1Aa (Figure 3C), consistent with previous reports [19,26,30,33]. Sf9 cells expressing EGFP alone did not swell upon their exposure to 5000 nM Cry9Aa (Figure 3D).

The Sf9 cells swelled when exposed to >50 nM Cry1Aa and Cry8Ca as well as >500 nM Cry1Da (Figure 4A). When exposed to these toxins at 5000 nM, approximately 10% of cells swelled with Cry1Aa, 7% with Cry1Da, and 9% with Cry8Ca. The percentage of swollen cells increased in a dose-dependent manner (Figure 4B). However, Cry9Aa at 5000 nM did not cause cell swelling (Figure 4A). These results imply that BmABCC1 functions as a receptor for Cry9Aa and that BmABCC4 functions as a receptor for Cry1Aa, Cry1Da, and Cry8Ca. The Sf9 cells expressing EGFP alone did not swell upon their exposure to 5000 nM Cry1Aa, Cry1Da, and Cry8Ca (Figure 4C). Thus, the Sf9 cell swelling assays confirmed the results obtained with HEK293T cells (Figure 3).

### 3.4. Binding Affinities of Cry1Aa, Cry1Da, Cry8Ca, and Cry9Aa to BmABCC1 and BmABCC4

We used SPR to analyze the binding affinities of Cry1Aa, Cry1Da, Cry8Ca, and Cry9Aa to BmABCC1 and BmABCC4. The Cry1Aa and Cry1Da sensorgrams indicated fast binding and fast dissociation rates with respect to BmABCC1, whereas the Cry8Ca and Cry9Aa sensorgrams indicated fast binding and slow dissociation rates (Figure 5A; *K_D_* values Cry1Aa, 4.61 × 10^−5^ M; Cry1Da, 4.15 × 10^−7^ M; Cry8Ca, 3.48 × 10^−8^ M; and Cry9Aa, 1.89 × 10^−9^ M). Therefore, Cry9Aa had approximately 24,000-, 210-, and 18-fold higher binding affinities for BmABCC1 compared with Cry1Aa, Cry1Da, and Cry8Ca, respectively (Table 2).

The Cry1Aa, Cry1Da, and Cry8Ca sensorgrams showed slow binding and fast dissociation rates with respect to BmABCC4, whereas Cry9Aa showed negligible binding (Figure 5B; *K_D_* values Cry1Aa, 2.67 × 10^−7^ M; Cry1Da, 6.37 × 10^−8^ M; and Cry8Ca, 6.17 × 10^−8^ M). The sensorgram of Cry9Aa binding to BmABCC4 was similar to the sensorgram of BSA binding to BmABCC2 (Figure 5B,C). This allowed us to verify the non-specific binding of proteins that do not bind to ABC transporters. Cry1Aa, Cry1Da, and Cry8Ca exhibited similar binding affinities for BmABCC4, whereas the *K_D_* value of Cry9Aa for BmABCC4 was not determined. There were weak correlations between the binding affinities of the Cry toxins for BmABCCs and their functions as receptors in cultured cells (Figure 3, Figure 4 and Figure 5).

### 3.5. Susceptibilities of BmABCC1- and BmABCC4-Knockout Silkworm Strains to Cry Toxins

We generated two silkworm strains—C1T09 with a 14-base deletion in the extracellular loop 1 (ECL1) region of BmABCC1 (Appendix A), and C4T02 with a 5-base deletion in the transmembrane domain 1 (TM1) of BmABCC4 (Appendix A). We subjected C1T09, C4T02, and wild-type silkworms to diet overlay bioassays. We calculated the 50% lethal concentration (LC_50_) and 95% confidence interval for each Cry toxin and analyzed resistance ratios through division by the LC_50_ values of wild-type silkworms (Figure 6, Table 3).

For Cry9Aa, the LC_50_ values of C1T09 and the wild type were 0.8 and 1.1 ppm, respectively. The resistance ratio for C1T09 was 0.8 (Figure 6A, Table 3). These findings suggest that, whereas Cry9Aa exhibits cytotoxicity to BmABCC1-expressing cultured cells, BmABCC1 is not a determinant of Cry9Aa sensitivity in silkworms. For Cry1Da, Cry8Ca, and Cry9Aa, the LC_50_ values of C4T02 and the wild type were 0.5 and 1.0, 37.3 and 28.6, and 0.5 and 0.4 ppm, respectively (Figure 6B, Table 3). The resistance ratios of C4T02 were 0.6-, 1.3-, and 1.1-fold for Cry1Da, Cry8Ca, and Cry9Aa, respectively (Figure 6B, Table 3). Therefore, whereas Cry1Da and Cry8Ca exhibit cytotoxicity to BmABCC4-expressing cultured cells, BmABCC4 is not a determinant of sensitivity to these Cry toxins in silkworms.

## 4. Discussion

### 4.1. ABC Transporters as Functional Receptors for Cry Toxins

Although ABC transporters are important receptors even for phylogenetically distant types of Cry toxins [12,13,14,15], the receptors for many Cry toxins are unknown. In this study, we analyzed the expression patterns of ABC transporters in the silkworm midgut. Forty-six ABC transporter genes showed expression levels of E-02 to E+01 (Appendix A). The Cry toxin susceptibility determinants BmABCA2, BmABCC2, BmABCB1, and BmABCC3 were highly expressed.

Fourteen combinations, comprising six BmABCCs and five Cry toxins, caused swelling in HEK293T cells (Figure 2). In the validation assays with Sf9 cells, the combinations of BmABCC1 with Cry9Aa and of BmABCC4 with Cry1Aa, Cry1Da, and Cry8Ca caused swelling (Figure 3 and Figure 4). However, even at 5000 nM, only 10–15% of Sf9 cells began to swell when exposed to the combination of BmABCC1 with Cry9Aa and the combination of BmABCC4 with Cry1Aa, Cry1Da, and Cry8Ca. This was lower than the percentage of cell swelling upon exposure to Cry1Aa in BmABCC2-expressing Sf9 cells, which approached 90% when 10 nM Cry1Aa was used [24]. Therefore, BmABCC1 and BmABCC4 are low-efficiency receptors with limited abilities to induce Cry toxin-mediated pore formation in the plasma membrane.

A receptor for Cry toxins must have high binding affinities for such toxins. The SPR analysis revealed that Cry9Aa, which uses BmABCC1 as a low-efficiency receptor, has high binding affinity for BmABCC1. Moreover, Cry1Aa, Cry1Da, and Cry8Ca, which use BmABCC4 as a low-efficiency receptor, have high binding affinities for BmABCC4 (Figure 5, Table 2). The *K_D_* values for the binding between BmABCCs and Cry toxins, together with the minimum concentrations of Cry toxins required to induce swelling in BmABCC-expressing Sf9 cells, are listed in Table 4 [19,20,24,34]. Cry toxins and BmABCCs with *K_D_* values < 10^−6^ M may show a receptor function in a heterologous cell expression system in which cell swelling can be observed (Table 4). These findings are consistent with previous analyses of the effects of Cry1Aa mutants on BmABCC2-expressing Sf9 cells [22,30].

BmABCC1 had a high binding affinity for Cry8Ca (3.48 × 10^−8^ M) but no receptor function (Table 3 and Table 4). Cry1Aa and Cry1Ab had *K_D_* values of ~10^−8^ M, but the minimum concentrations required to induce swelling in the BmABCC3-expressing Sf9 cells were 1 nM for Cry1Aa and 1000 nM for Cry1Ab; this constituted a 1000-fold difference (Table 4) [19,34]. These results suggest that a high binding affinity does not necessarily translate to a high cytotoxicity, and the *K_D_* values were not fully aligned with the minimum concentrations required to initiate cell swelling [22,30]. These results are consistent with findings concerning Cry toxin mutants, among which a high receptor-binding affinity is not correlated with cytotoxicity [26,30,35]. Thus, cytotoxicity is likely dependent on factors other than the binding affinity between the receptor and the Cry toxin.

The ECL1 and ECL4 regions of ABCC2 and ABCC3 are involved in the binding to Cry1A and Cry1F [21,31,36,37]. Therefore, it is considered that the ECLs of ABC transporters function as toxin-binding sites. However, sequence alignments revealed that the amino acid sequences of ECLs 1 to 6 exhibit a low homology, with significant differences in loop length, particularly concerning ECL4 (Appendix A). On the other hand, the domain II loop region of Cry1Aa is involved in binding to BmABCC2 [23,30], so the domain II loop region of Cry toxins is considered to function as an ABC transporter binding site. However, the lengths and amino acid sequences of the loops differ among the four Cry toxins. Boström and Schmitt published a comparative study on protein binding, in which they extracted data from the Protein Data Bank (PDB), a database of protein crystal structures, to determine how ligands with similar structures bind to the same protein [37]. They indicated that ligands with similar sequences or structures are more likely to occupy the same binding pocket in a similar manner [37]. In the various combinations of BmABCCs and Cry toxins demonstrated in the present study, the sequences and structures of the regions of the putative binding sites were diverse. Binding in common regions despite differences in sequences or structures deviates from the general rules of protein binding. Therefore, we speculate that there may be a unique binding mechanism between ABC transporters and Cry toxins.

The loop region of domain II in Cry1Aa binds to BmABCC2 and to a cadherin-like protein, BtR175 [30]. The loop regions of Cry toxins have a long groove structure that exhibits flexibility through its intrinsic loop characteristics. Therefore, we hypothesized that Cry1Aa achieves promiscuous binding by offering different amino acid residues at distinct positions as critical binding points for receptor binding [30]. This property presumably enables Cry toxins to bind ABC transporters from various lineages. Based on our screening in HEK293T cells (Figure 2) and evaluation of binding (Figure 5), we overlaid the relationships of BmABCC–Cry toxin pairs with receptor functions onto the phylogenetic tree of ABCC transporters (Figure 7). One Cry toxin could bind to multiple members of diverse BmABCC lineages, spanning a considerable phylogenetic distance but exhibiting a low receptor functionality (Figure 7, Table 4). For example, the ability of Cry9Aa to bind to four phylogenetically distant BmABCCs (BmABCC1, 4, 10, 11) may depend on the above-mentioned binding characteristics of Cry toxins. We investigated only six ABC transporters; the forty-six ABC transporters expressed in the silkworm midgut could include low-efficiency or highly efficient receptors beyond the “C” subfamily of ABC transporters.

### 4.2. Importance of Low-Efficiency Receptors That Are Not Cry Toxin Susceptibility Determinants

The *K_D_* value of BmABCC1 for Cry9Aa was 1.89 × 10^−9^ M (Table 2), which is relatively high compared to the series of relationships in prior reports (Table 4). Cry9Aa was cytotoxic to BmABCC1-expressing Sf9 cells only at >500 nM (Figure 3A, Table 4). The BmABCC1 knockout strain did not show resistance to Cry9Aa (Table 3). Therefore, BmABCC1 is a low-efficiency receptor for Cry9Aa but is not a susceptibility determinant, possibly because the midgut of silkworm larvae exhibits other highly efficient receptors with higher receptor functions for Cry9Aa compared to BmABCC1.

In a heterologous expression system involving Sf9 cells, BmABCC4 showed low receptor functions for Cry1Da and Cry8Ca (Figure 4, Table 4). Consistent with these findings, the C4T02 strain did not show resistance to Cry1Da or Cry8Ca (Figure 6, Table 3). Thus, BmABCC4 may not be a determinant of silkworm susceptibility to Cry1Da and Cry8Ca, possibly due to the presence of highly efficient receptors which confer a higher susceptibility compared to BmABCC4. Indeed, Cry1Ac can induce toxicity in silkworm via the synergistic receptor functions of BmABCC2 and BtR175 and BmABCC3 and BtR175 [19]. However, the former combination exhibits a greater potential for toxicity. Knockout of BmABCC3 did not result in Cry1Ac resistance in silkworm and, apparently, caused no change, since the synergistic receptor function of BmABCC2 and BtR175 still remained [19].

This study showed that not only BmABCC2 and BmABCC3 but also BmABCC4 function as receptors for Cry1Aa (Figure 2). BmABCC2 and BmABCC3, highly efficient receptors for Cry1Aa, play nearly equivalent roles as susceptibility determinants because of their synergism with BtR175 [34]. Furthermore, the double knockout of BmABCC2 and BmABCC3 decreased the susceptibility by >100,000-fold, indicating that BmABCCs other than BmABCC2 and BmABCC3 are not involved in Cry1Aa susceptibility [34]. Therefore, we did not perform a diet overlay bioassay of Cry1Aa in the BmABCC4-knockout strain. BmABCC4 is presumably a low-efficiency receptor for Cry1Aa in silkworm.

The reason for Cry toxins retaining various ABC transporters as low-efficiency receptors that do not participate in susceptibility determination may be to establish quickly receptor relationships that enable a more efficient induction of cytotoxicity when circumstances change. Combinations of mutations may allow Cry toxins to bind with a high affinity to ABC transporters, resulting in toxicity to insects. In other words, at a *K_D_* value of <10^−6^ M, they begin to function as low-efficiency receptors, whereas, at *K_D_* values of around 10^−8^–10^−10^ M, the ABC transporter may act as a highly efficient receptor; such receptors could determine the susceptibility of insects to Cry toxins (Table 4). For example, the *K_D_* values of BmABCC2 and BmABCC3, which are susceptibility determinants for Cry1Aa, are 3.1 × 10^−10^ and 3.4 × 10^−8^ M, respectively [20,24], whereas the *K_D_* value of the low-efficiency receptor BmABCC4 is 2.7 × 10^−7^ M (Table 2). The accumulation of mutations may enable Cry toxins to use low-efficiency receptors as highly efficient receptors. The amino acid sequences of ABC transporters display variation among insect species, even for orthologues. This sequence variation in ABC transporters potentially enables them to serve as higher-efficiency receptors for existing toxins. In summary, the numerous low-efficiency receptors observed imply a basis for the evolution of Cry toxins. The existing relationship between Cry toxins and ABC transporters that serve as Cry toxin susceptibility determinants is likely the result of selection among receptors that exhibited low-efficiency relationships with Cry toxins. The interactions of Cry toxins with low-efficiency receptors are presumably facilitated by the promiscuous binding property of their loop regions, which provides numerous opportunities for binding to diverse molecules. The ability of Cry toxins to bind to multiple ABC transporters—in other words, to have multiple potential receptors—likely promoted their adaptation to particular target cells and insect taxa. It is this promiscuous binding property that drove the evolution of Cry toxin subfamily members that bind to ABC transporter subfamily members. As a result of this evolution, toxins like the Lepidopteran-active Cry1Aa, Dipteran-active Cry2Aa, and Coleopteran-active Cry1Ba, Cry1Ia, Cry9Da, and Cry3Aa, which are phylogenetically distant, use ABCC2, ABCA2, and ABCB1 as highly efficient receptors beyond the ABC transporter subfamily group [6,12,13,14,15,16].

## 5. Conclusions

A cell swelling assay using a heterologous expression system in HEK293T or Sf9 cells revealed that five Cry toxins use multiple BmABCCs as low-efficiency receptors, which induce cytotoxicity only at high concentrations. We hypothesize that promiscuous binding property of Cry toxins’ receptor-binding region enables them to bind to diverse ABC transporters. Such diversity likely drove the evolution of Cry toxin subfamily members that bind to ABC transporter subfamily members.

## Figures and Tables

**Figure 1 biomolecules-14-00271-f001:**
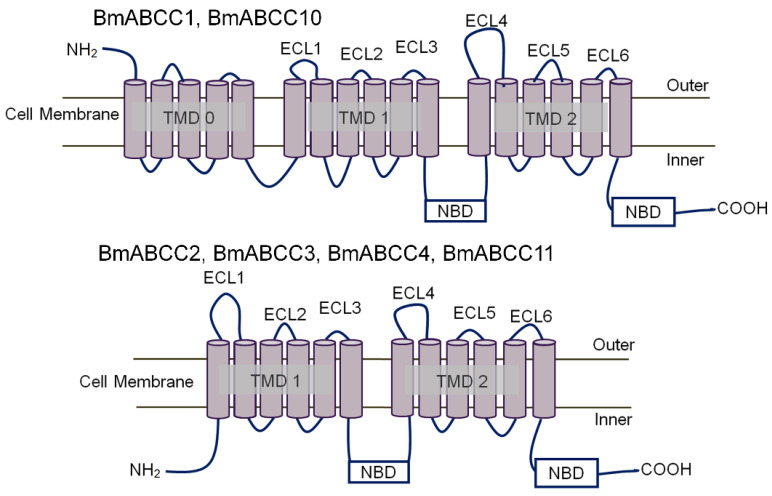
Predicted topologies of BmABCCs. Predicted topologies of the six BmABCCs used in this study generated using PolyPhobius (https://phobius.sbc.su.se/poly.html, accessed on 10 December 2023). TMD, transmembrane domain; NBD, nucleotide-binding domain; and ECLs, extracellular loops.

**Figure 2 biomolecules-14-00271-f002:**
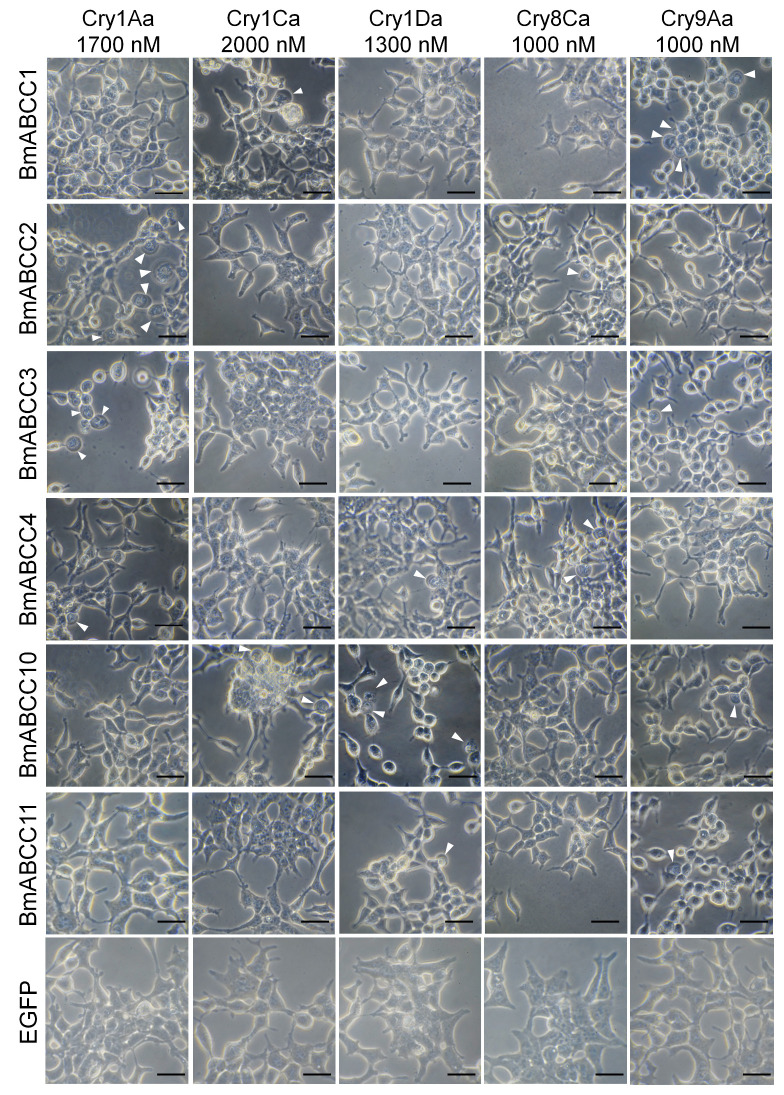
Cry toxin susceptibilities of HEK293T cells expressing six BmABCCs. Cells expressing BmABCCs were incubated with Cry toxins at 37 °C for 1 h and observed by phase-contrast microscopy. Arrowheads, swollen cells. Scale bar = 20 μm.

**Figure 3 biomolecules-14-00271-f003:**
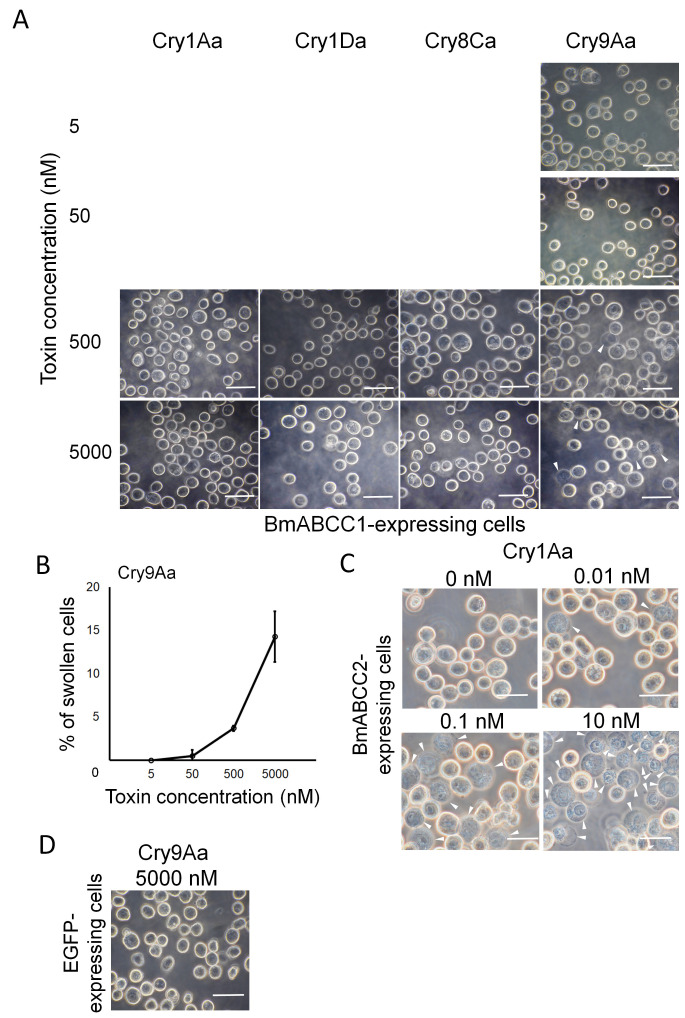
Induction of Cry toxin-mediated cell swelling by BmABCC. (**A**,**B**) BmABCC1-expressing Sf9 cells were incubated for 1 h with phosphate-buffered saline (PBS) containing Cry1Aa, Cry1Da, Cry8Ca, and Cry9Aa. (**A**) Swollen cells with low refraction (arrowhead) observed by phase-contrast microscopy. Scale bars = 20 μm. (**B**) Cry9Aa dose–response curve of BmABCC1-expressing Sf9 cells. Each experiment involved counting over 300 cells (about 20 fields under the microscope) at a single concentration to calculate the swollen cell rate. The points represent the average of the results obtained via two assays. (**C**) BmABCC2-expressing Sf9 cells were incubated for 1 h with phosphate-buffered saline (PBS) containing Cry1Aa. (**A**,**C**) Swollen cells with low refraction (arrowhead) that expressed BmABCC1 or BmABCC2 were observed by phase-contrast microscopy. Scale bars = 20 μm. (**D**) EGFP-expressing Sf9 cells were incubated for 1 h with PBS containing 5000 nM Cry9Aa and then observed by phase-contrast microscope. Scale bars = 20 μm.

**Figure 4 biomolecules-14-00271-f004:**
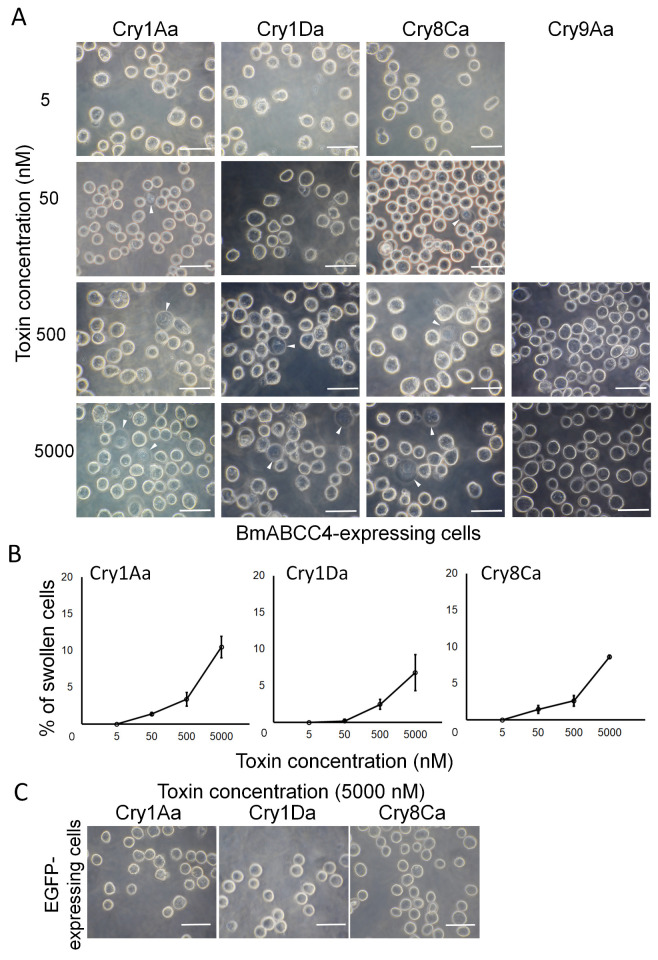
Induction of Cry toxin-mediated cell swelling by BmABCC4. BmABCC4-expressing Sf9 cells were incubated for 1 h with PBS containing Cry1Aa, Cry1Da, Cry8Ca, and Cry9Aa. (**A**) Swollen cells with low refraction (arrowhead) that expressed BmABCC4 observed by phase-contrast microscope. Scale bars = 20 μm. (**B**) Cry toxin dose–response curves of BmABCC4-expressing Sf9 cells. Each experiment involved counting over 300 cells (about 20 fields under the microscope) at a single concentration to calculate the swollen cell rate. The points represent the average of the results obtained via two assays. (**C**) EGFP-expressing Sf9 cells were incubated for 1 h with PBS containing 5000 nM Cry1Aa, Cry1Da, and Cry8Ca and then observed by phase-contrast microscopy. Scale bars = 20 μm.

**Figure 5 biomolecules-14-00271-f005:**
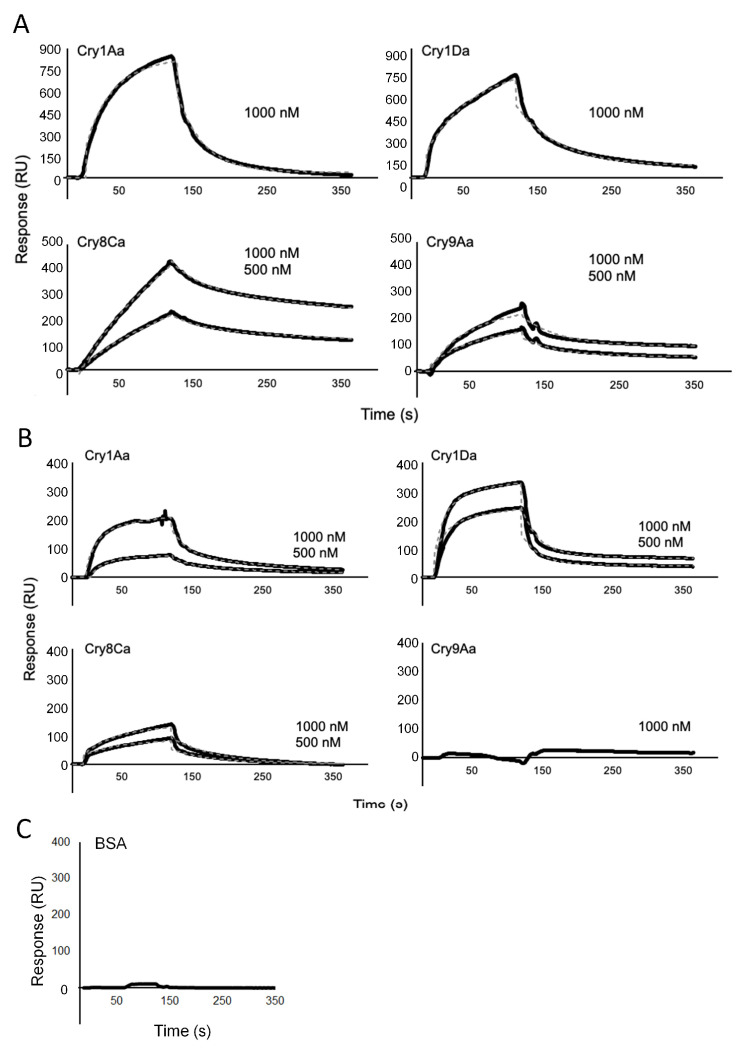
SPR analysis of the binding of Cry1Aa, Cry1Da, Cry8Ca, and Cry9Aa to BmABCC1 and BmABCC4. (**A**,**B**) Sensorgrams of Cry1Aa, Cry1Da, Cry8Ca, and Cry9Aa binding to BmABCC1 (**A**) and BmABCC4 (**B**). (**C**) Sensorgram of BSA binding to BmABCC2. In the analysis, the BmABCC2 purified in previous studies was used [30]. (**A**–**C**) Responses were recorded for 120 s during the association phase and 240 s during the dissociation phase. Black solid lines are the measured response curves; the gray dashed lines are the fitted curves drawn with the BIACORE accompanying software, BIAevaluation ver.4.1, based on two-state reaction models.

**Figure 6 biomolecules-14-00271-f006:**
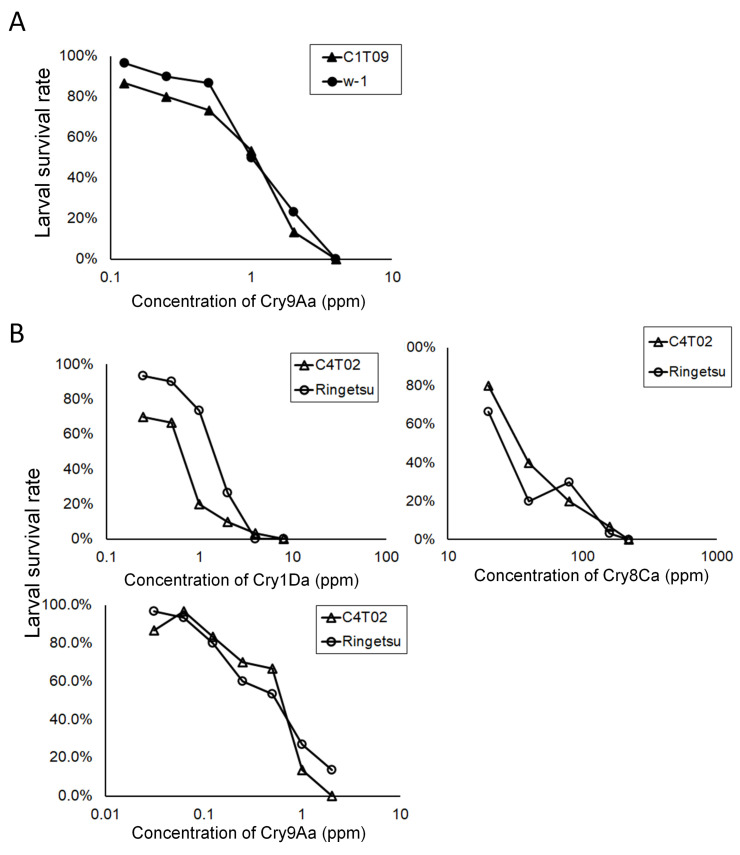
Concentration–response curves of a diet overlay bioassay to evaluate the susceptibility of *B. mori* larvae. (**A**) Diet overlay bioassay using Cry9Aa with second-instar larvae of the knockout strains. C1T09, a BmABCC1 knockout strain, and the wild-type strain (w-1) were evaluated. (**B**) Diet overlay bioassay using Cry1Da, Cry8Ca, and Cry9Aa with second-instar larvae of the knockout strains. C4T02, a BmABCC4 knockout strain, and the wild-type strain (Ringetsu) were analyzed.

**Figure 7 biomolecules-14-00271-f007:**
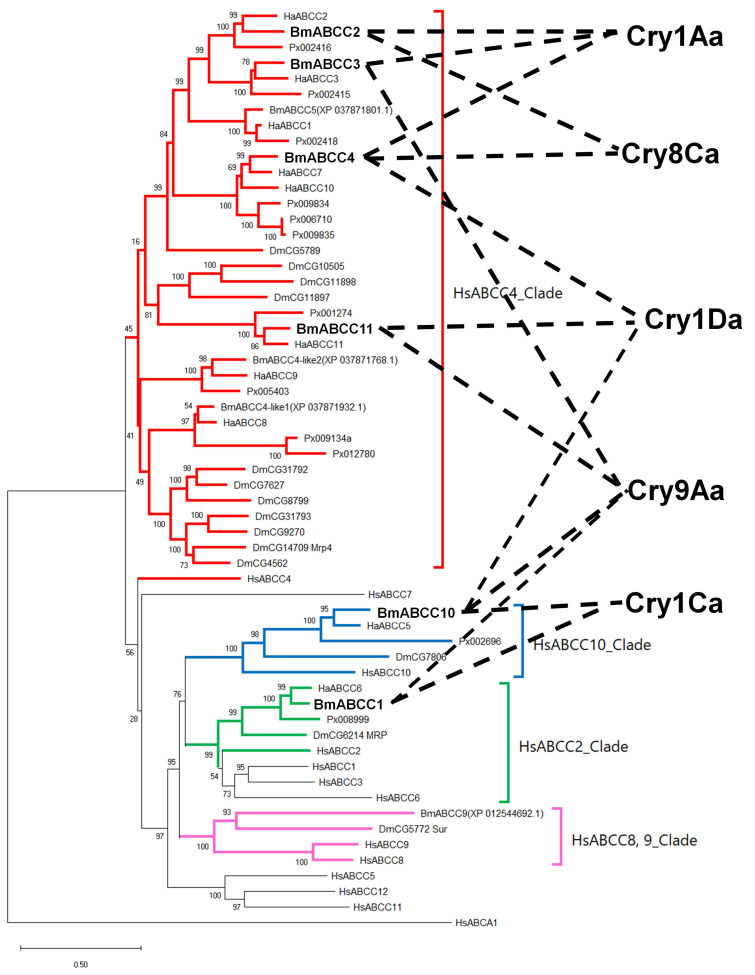
ABCCs that function as receptors for Cry toxins. Based on the Cry toxin susceptibilities of HEK293T cells expressing BmABCCs, we identified ABCCs that function as Cry toxin receptors in the context of the phylogenetic tree in Appendix A. The BmABCCs and their receptor Cry toxins are indicated by dashed lines.

**Table 1 biomolecules-14-00271-t001:** BLAST analysis results of the six BmABCCs used in this study.

BmABCCs Used in This Study		Most Closely Related Protein in *H. sapiens*		Most Closely Related Protein Except for *Bombyx* sp.	
Name in This Study (Name in the NCBI)	Protein ID	Name in NCBI	Protein ID (Max Score ^a^)	Name (Species)	Protein ID (Max Score ^a^)
BmABCC1 (MRP1)	XP_012546136.1	MRP1 (ABCC1)	NP_004987.2 (1476)	MRP1 (*Manduca sexta*)	XP_030027498.1 (2743)
BmABCC2 (ABCC2)	NP_001243945.1	ABC transporter MOAT-B (ABCC4)	AAC27076.1 (919)	MRP4 (*M. sexta*)	XP_037300509.1 (2061)
BmABCC3 (ABC)	XP_037872082.1	ABCC4	KAI2569693.1 (947)	MRP4 (*M. sexta*)	XP_030036633.2 (2253)
BmABCC4 (MRP4)	XP_021205339.2	ABCC4	KAI2569693.1 (985)	ABCC4 (*Spodoptera frugiperda*)	XP_035429631.1 (2207)
BmABCC10 (MRP7)	XP_004925578.1	ABCC10	XP_047275455.1 (821)	MRP7 (*Hyposmocoma kahamanoa*)	XP_026315550.1 (2427)
BmABCC11 (ABC-lethal)	XP_037875232.1	ABCC4	KAI2569693.1 (901)	MRP-lethal (*M. sexta*)	XP_030028866.1 (2335)

**^a^** Max score is the highest alignment score calculated in the BLAST analysis.

**Table 2 biomolecules-14-00271-t002:** *K_D_* values of Cry toxins for binding to BmABCC1 and BmABCC4.

	Cry1Aa	Cry1Da	Cry8Ca	Cry9Aa
BmABCC1	4.61 × 10^−5^ M	4.15 × 10^−7^ M	3.48 × 10^−8^ M	1.89 × 10^−9^ M
BmABCC4	2.67 × 10^−7^ M	6.37 × 10^−8^ M	6.17 × 10^−8^ M	Not detectable

**Table 3 biomolecules-14-00271-t003:** Susceptibilities of knockout and wild-type strains to Cry toxins.

	*B. mori* Strains	N ^a^	Cry Toxin LC_50_ (ppm, 95%CI) ^b^	Slope ^c^	RR ^d^
Cry1Da	C4T02	210	0.5 (0.39~0.69)	2.2	0.6
	Wild-type	210	1.0 (0.72~1.25)	1.9	1.0
Cry8Ca	C4T02	180	37.3 (27.93~47.06)	2.6	1.3
	Wild-type	180	28.6 (18.81~37.47)	2.3	1.0
Cry9Aa	C4T02	240	0.5 (0.36~0.66)	1.6	1.1
	Wild-type	240	0.4 (0.31~0.58)	2.1	1.0
Cry9Aa	C1T09	240	0.8 (0.60~1.17)	1.7	0.8
	Wild-type	240	1.1 (0.82~1.45)	2.3	1.0

^a^ Number of larvae tested. ^b^ Concentration of toxins killing 50% of larvae and its 95% confidence interval (CI). ^c^ Slope of the concentration–mortality line. ^d^ Resistance ratio (RR) = LC_50_ of knockout strain divided by LC_50_ of the same toxin for wild-type. C4T02 is a knockout strain for BmABCC4; C1T09 is a knockout strain for BmABCC1.

**Table 4 biomolecules-14-00271-t004:** Partial correlations of Cry toxin-binding affinity and Sf9 cell swelling activity.

Receptor	Toxin	*K_D_* (M) ^a^	Effective Conc. (nM) in Sf9 Cell Swelling Assay ^b^
BmABCC1	Cry1Aa	4.6 × 10^−5^	>5000
	Cry1Da	4.2 × 10^−7^	>5000
	Cry8Ca	3.5 × 10^−8^	>5000
	Cry9Aa	1.9 × 10^−9^	500
BmABCC2	Cry1Aa	^e^ 3.1 × 10^−10^	^e^ 0.1
	Cry1Ab	^c^ 5.2 × 10^−9^	^e^ 10
	Cry1Ac	^c^ 2.3 × 10^−10^	^e^ 1
	Cry1Fa	^c^ 2.0 × 10^−10^	^c^ 1
BmABCC3	Cry1Aa	^d^ 3.4 × 10^−8^	^f^ 1
	Cry1Ab	^c^ 4.4 × 10^−8^	^c^ 1000
	Cry1Ac	^c^ 8.1 × 10^−8^	^c^ 1000
	Cry1Fa	^f^ 3.9 × 10^−9^	^f^ 10
BmABCC4	Cry1Aa	2.7 × 10^−7^	50
	Cry1Da	6.4 × 10^−8^	500
	Cry8Ca	6.2 × 10^−8^	50
	Cry9Aa	Not detectable	>5000

^a^ The values of *K_D_* were cited from ^c^ Wang et al. (2021) [19] and ^d^ Endo et al. (2018) [20]. ^b^ The lowest concentrations of toxins by which the swelling of Sf9 cells was obviously induced were sourced from ^c^ Wang et al. (2021) [19] and reported from ^e^ Tanaka et al. (2013) [24] and ^f^ Kim et al. (2022) [34].

## Data Availability

The original contributions presented in the study are included in the article/Appendix A, further inquiries can be directed to the corresponding author/s.

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
