# Peer review of "Cry Toxins Use Multiple ATP-Binding Cassette Transporter Subfamily C Members as Low-Efficiency Receptors in Bombyx mori"

_biomolecules, 2024, doi:10.3390/biom14030271_

Round 1

Reviewer 1 Report (Previous Reviewer 3)

Comments and Suggestions for Authors

The reviewer found that the re-submitted manuscript has been improved significantly in terms of clarity and scientific soundness.

Minor point

1. Line 202-203, there are two consecutive “however”. 

Comments on the Quality of English Language

There are minor text errors that can probably be fixed during proofreading. 

Author Response

Thank you for your carefully reviewing. We removed redundant "however" in Line 203.

Reviewer 2 Report (Previous Reviewer 1)

Comments and Suggestions for Authors

Some images are missed and should be supplemented. Also, the fluorescence signal in Fig3D and Fig4C should be exhibited.

Author Response

Thank you for your suggestion. We added the fluorescence image regarding to Fig. 3C (BmABCC2-expressing Sf9 cells) in Fig. S7. We added some explanation about them to the manuscript (Lines 270-271,537-538). On the other hands, the fluorescence image regarding to Fig. 3D and Fig. 4C (BmABCC1- and BmABCC4-expressing Sf9 cells) were already showed in Fig. S7, so there is no additional figures about them.

Round 2

Reviewer 2 Report (Previous Reviewer 1)

Comments and Suggestions for Authors

The authors have responded to the reviewers' comments well. The current version should be considered to publish.

This manuscript is a resubmission of an earlier submission. The following is a list of the peer review reports and author responses from that submission.

Round 1

Reviewer 1 Report

Comments and Suggestions for Authors

The experimental design is too simple and no further investigation regarding mechanism is carried out. The present data is inadequate for publication. 

Comments on the Quality of English Language

Minor editing of English language required.

Reviewer 2 Report

Comments and Suggestions for Authors

This is an interesting and elegant paper describing the role of the ABC transporters subfamilty C as potential receptors of the Cry toxins. The paper is well written and structured, results are relatively clearly present. Discussion is comprehensive and relevant to the topic of the paper. The main finding of the work is that ABC transporters can serve as the receptors for the Cry toxins though the efficiency of interacting is relatively low.

I have several comments regarding the manuscript:

1.    Probably, “low efficient” term would be better that “low functional” in the title and over the text.

2.    The most important issue regarding the experimental part of the paper is that the negative controls are probably missed for experiments shown in Figures 1-3.

3.    When comparing the efficiency of Cry toxin interacting by the ABC transporters an additional control with any protein that does not bind to ABC transporters would be useful to check for non-specific binding.

4.    In my opinion, Table 1 can be moved to the Supplement and be substituted, for example, with the Figure illustrating structures of the selected transporters.

Comments on the Quality of English Language

Minor correction of English would improve the Manuscript. 

Reviewer 3 Report

Comments and Suggestions for Authors

In this study, the authors analyzed the transcription level of ABC transporters in the midgut of the silkworm. They selected members of the ABCC family to study whether these are receptors for cry toxins. The authors used the cell swelling assay in HEK cells expressing different silkworm ABCCs to screen for potential receptors for five cry toxins. Then they further validated the targets in sf9 cells by SPR assay using purified proteins. Finally, the authors generated knockout strains of BmABCC1 and BmABCC4 and tested their susceptibility to the identified cry toxin binders from the cell-based assays and SPR. They found that these two ABCCs are not the determinant of susceptibility in silkworms to the tested cry toxins.

The manuscript is well written and the methods and results are presented clearly. However, the reviewer has a few comments.

Major points

1. When selecting potential targets of BmABCCs for the study, the authors used the transcription per million as an indicator of expression level. In the discussion session, Line 337-338, the authors claimed that “Therefore, ABC transporters with high expression levels are high-functional 343 receptors for Cry toxins”. The reviewer thinks that it is an overstatement. The fact that four of the known susceptibility determinants are highly expressed does not necessarily mean that all susceptibility determinants need to have high expression levels.

2. The SDS-PAGE in Figure S1 does not look right. Multiple papers on the biochemistry and structure of ABCCs have been published. On SDS-PAGE, the protein typically shows up as a monomer at around 150-200 kDa. However, in this study, the band is around 300 kDa. The authors explained it as a dimer. However, this either indicates that the SDS-PAGE was not run properly or that the band may not be the presumed ABCC1 or ABCC4. The authors did not conduct a western blot to verify the identity of the band or even the presence of the Flag tag in the 300 kDa band. Without this evidence, it is unclear whether the protein was successfully purified and it affects both the credibility and the interpretation of the SPR results.

3. In the cell swelling assays, the reviewer recommends that the authors include positive control using established ABC transporter and cry toxin pairs and negative control using un-transfected or un-infected cells. It would be better to also mark the swelled cells in the fluorescent microscope images in the supplementary figures.

4. Figure 2, panel B. The points in the curve do not have error bars. There is no description of how the data points were obtained. For example, from how many images and how many repeats of cell culture.

5. Figure 5, in the cry8ca figure, the legends and the data points do not match.

Minor points

1. In the method session, line 185, “I added ….”. Please rephrase the sentence.

2. Line 269, the authors should specify that the sf9 cells here are expressing BmABCC4.

3. Line 375-376. Please change “transmembrane domains” to “transmembrane helixes”. The use of the word “comprising” here is a bit strange. ECLs and transmembrane helixes are different components of the protein.